# Agritourism and Farms Diversification in Italy: What Have We Learnt from COVID-19?

Barbara Zanetti [†], Milena Verrascina [†], Francesco Licciardo [†] and Giuseppe Gargano *,[†]

Council for Agricultural Research and Economics, Research Centre for Agricultural Policies and Bio-Economy, 00187 Rome, Italy; barbara.zanetti@crea.gov.it (B.Z.); milena.verrascina@crea.gov.it (M.V.); francesco.licciardo@crea.gov.it (F.L.)
* Correspondence: giuseppe.gargano@crea.gov.it
† The authors contributed equally as co-first authors.

**Abstract:** The COVID-19 pandemic has rapidly disrupted the household behavior in all areas and also those related to eating and daily food. Research carried out shows there have been significant changes compared to pre-COVID levels in the way consumers plan their food purchases. Based on the results of empirical data and emerging information such as ad hoc reports and analysis of academic literature, the authors aim to understand the effect of COVID-19 on agricultural and extra-agricultural activities in diversified Italian farms. More specifically, due to their importance at a national level, the focus of the analysis is represented by the agritourism, how they have reacted to the challenges posed by the pandemic, and towards which evolutionary lines they are orienting themselves to face the next future challenges. Empirical data for this study were collected through the use of a questionnaire survey, managed by the research team. The survey, conducted online during summer 2021, was designed by using a random stratified sampling for which the farms are characterized by a certain heterogeneity of the activities carried out (i.e., hospitality, processing of products, renewable energy production, etc.). The research activity covered the entire Italian territory and the number of responding farms with agritourism activities is equal to 77 (a 17.5% response rate). The results highlight the importance of farm with agritourism activities in dealing with COVID-19 crisis and policy implications in terms of support for the competitiveness of farms, exchange of knowledge, and innovations among farmers that should be taken into consideration to target the next rural development policy at the EU, the national and regional level. At the same time, the sample reaction methods to the pandemic and the changing business strategies highlight a certain resilience of Italian farms with agritourism activities, thus showing their ability to adapt.

**Keywords:** COVID-19 crisis; resilience; farm diversification; agritourism; sustainable local food system

## 1. Introduction

The Coronavirus Disease-2019 (COVID-19) pandemic has rapidly disrupted the household behavior in all areas, including those related to eating and daily food. If compared to previous ones, many believe that this pandemic caused the gravest problems not only in the public health sector but also in terms of economic and social security as well as the quality of life [1].

Research carried out [2] shows that there have been significant changes compared to pre-COVID levels in the way consumers plan their food purchases and deal with the technology of food storage in terms of at-home meal preparation and consumptions.

During the first weeks of the pandemic, national governments imposed quarantine conditions to limit the spread of Coronavirus [2] by forcing people to stay at home and to go out only for basic needs such as purchasing food, medical emergencies, and to work if remote working was not possible. Further restrictive measures in food supply chains [3] were also taken, which caused a decline in productivity, higher transportation costs, a

significant loss of income for farmers, and an increase of the vulnerable products' prices such as vegetables and fruit. The imposed widespread lockdown and the consequent restrictions on the movement of people brought to changes in purchasing preferences in favor of fresh and trustworthy local food and an increase of online purchases and home delivering where meals were prepared and eaten at home [4]. People reduced the number of visits to the shops, supporting smaller/local business and shopping online more than previously.

Beyond the dramatic effects, the pandemic has revealed strong business responsiveness and new opportunities. Although the closure of borders, the suspension of all activities except essential ones, and of all movements on the national territory have distorted habits, on the other hand, and in a few months, some economic, social, and environmental trends have strongly emerged which will probably draw new models and consolidate some opportunities in the rural areas. These measures influenced the structure of demand and the entrepreneurial activity and to reconsider the role of agriculture [5], how essential it is to improve the environment, and contributes to sustainable development based on fair, green and healthy agri-food systems.

It has indeed fostered awareness among people regarding the importance of agrofood products. Various studies [6–8] focusing on the changes on consumer behavior and preferences during the pandemic crisis showed that many people attempted to eat a healthier diet rich in fruit and vegetables [9]. According to Basev [10], consumer behavior is a complex process which involves several factors, including economic, regional diversity, and social, as well as preferences and attitudes, which all affect food consumption.

The agricultural sector has shown an ability to further reorganize and seize some new opportunities by diversifying proposals and innovating processes, looking with optimism at a future that is in many ways still uncertain. To react to the difficult context triggered by the lockdown, many farms immediately concentrated their work by reorganizing it where necessary and trying to identify new strategies to transform their products and for a closure of the supply chain by trying to reach the final consumer independently with direct sales, to overcome the logistical and organizational difficulties of the usual channels. These latter channels were strengthened and, in some cases, activated from scratch by the farms, which also oriented corporate communication and implemented the internal organization system. They activated the e-commerce service through their website or phone app and the home delivery service of fresh, processed products and—where permitted by regional regulations—of ready meals, especially on Sundays and holidays, with all that ensued in organizational terms (collection of orders, preparation of the delivery service and often of a dedicated and refrigerated vehicle, etc.) [11,12].

Many people decided to leave their usual homes for shorter or longer periods to spend the lockdown in extra-urban areas, near the sea or in the countryside where they can continue their work and educational, recreational, sport, and social activities aimed at families and individuals. Thanks to the internet, it was possible to continue working remotely, studying, and following lessons, to access information, health services, and public administration, to keep in touch with family, friends, and the rest of society. Currently, remote working is still one of the measures adopted to limit the spread of the Coronavirus, but once the emergency is over, an important portion of delocalized work will probably remain [13–16].

As a result of a new demand, some farms have partly reorganized the hospitality service and many plan to do so in the coming months, offering stays for long periods and spaces equipped for remote work of guests. This can represent an opportunity for rural and extra-urban areas, also triggering reflections on the need for digitization of these areas as well as more efficient connection networks.

The impacts and reactions to the crisis are the subject of the present research about farms that have made an important contribution to the codification of possible trajectories for the future.

More specifically, it focuses on the effects of the COVID-19 pandemic on some Italian diversified farms, with the aim to investigate the consequences of the health emergency, their reactions, and their needs in order to overcome the crisis and to explore whether the changes required can produce opportunities for their activities.

The research work investigates a sample of diversified farms of all around Italy through semi-structured interviews. The motivation behind this choice is to be traced back to the fifty years of Italian agricultural experience which, over time, has created the conditions to strengthen direct sales and encourage the development of further non-agricultural activities.

The starting point of the study is that diversified farms have shown greater resilience to the crisis and that can cope better with the pandemic than other specialized farms. The research objective aims to understand the following questions: What is the effect of COVID-19 on agricultural and non-agricultural activities? How does the Italian model of a multifunctional farm react to the challenges posed by the pandemic? On which evolutionary lines do multifunctional farms position themselves?

Thanks to the responses received, it was possible to represent the first reactions of the farms facing the sudden crisis in terms of prospects and possible trajectories for the near future, with regard to main trends, new reference markets, role of farms in the provision of personal services, evolution of the channels of direct sales.

The analysis carried out highlights a leap in quality of farms that had already started a path of diversification of activities and income. Already included in an 'alternative' agri-food system based on direct sales and catering, they have exploited, to their advantage, the experience acquired in personal contact with the consumer, rebuilding relational networks and consolidating the relationship of trust with an extended clientele. They have used the nearest catchment area—the cities. In the cities they have indeed found a market made up of consumers attentive to the quality of the products, in need of reassurance on the food production process. Development of enhanced, robust agri-food chains will probably require a fine, complementary balance between the current 'global' food supply practices and other 'local' trends.

The results highlight the importance of farm diversification in dealing with COVID-19 crisis and highlight interesting policy implications in terms of support for the competitiveness of farms, exchange of knowledge, and innovations among farmers that should be taken into consideration to target the next rural development policy at the EU, national, and regional level.

## 2. The Theoretical Approach: Farm Diversification as a form of Resilience Strategy

The literature on farm diversification represents a form of resilience where the capacity of turning local resources into creative innovations reveals that farms can adapt well in case of adversity, an advantage that can be considered crucial when experiencing any socio-economic crises [17,18].

Farm diversification is analyzed in the framework of multifunctional agriculture meant as the potential of agricultural enterprises to produce products and services other than food. It is the strategy that many farms adopted to meet the decreasing profit margins due to agricultural overproduction and new needs expressed by the society [19].

The farm income basis can be diversified by broadening, deepening and regrounding. The broadening includes activities connected to agricultural and rural resources such as recreational activities and landscape management; deepening refers to the integration of food processing in the supply chains; regrounding refers to the mobilization of farm resources as new forms of cost reduction [20–23].

Salvioni et al. [24] argue that farm diversification includes three areas: (1) agricultural diversification (e.g., selling a specific mix of products); (2) product differentiation (e.g., organic production or products of protected designation of origin); (3) non-agricultural diversification (e.g., recreational activities). The combination of the agricultural diversification with non-agricultural activities can be considered a typical resilience strategy of small European farms where products requiring processing are more typically marketed

through integrated supply chains and the marketing channels differ among diversified and more focused business. In diversified farms, producers have to manage multiple roles at the same time, such as the products into which to invest their time and resources and the marketing channels.

In this framework, the implementation of tourism activities is a form of diversification of agricultural activities that has accompanied European farming in the last decades and was born from the necessity to find ways to increase income with the awareness of the important role of the agriculture in the relationship between natural resources and rural areas [25–27].

This has led to a rethinking of the traditional agricultural activities that continue to represent the main economic activity, especially in the most marginalized areas, even if they are no longer carried out exclusively [28]. Non-agricultural activities are becoming increasingly important in marginal rural areas where the possibilities to develop job opportunities are very limited or where the cultural and environmental heritage is very appreciated by tourists [29]. These activities are manifested through the presence of leisure and recreational services such as hospitality and dining and through the preservation and valorization of the territory and its products such as direct selling, soil conservation, ecosystem service conservation, and attention to the preservation of landscapes and biodiversity.

Agritourism farm has always been an adaptive tourism sector and quite agile in devising diverse solutions when experiencing any socio-economic crises.

There is a wide and consolidated literature on the economic and social benefits of the non-agricultural activities carried out by agritourism that shows positive performances, where attention of the farmer to environmental aspects and care of the territory is a consequence of a greater demand for a variety of products and traditional agricultural landscapes [30–33].

Moreover, several pre-COVID-19 studies conducted on agritourism [34–36] as a more complex diversification activity rooted in business strategies discovered that, in order to ensure sustainable development in rural areas, it had to be innovative and highly competitive.

There is a various interpretation of innovation, but it generally refers to the development of original products and marketing based on local culture [37]. A study on Italian farms carried out by Palmi and Lezzi [38] found that innovation of agritourism products consists of both tangible and intangible local traditions in the community and territory. These traditions are sourced from the farm products, the landscape, the historical events, and the heritage. The combinations of these sources lead to successful innovation in farm management, improved products, and attractive tourism marketing.

A definition of agritourism [19,23,24,39] can be found in many places of the world and concerns a wide variety of forms of rural tourism. Rural tourism is considered by the literature [40] as a phenomenon for which economic, social, and environment effects for population and territories depend on the connections between tourism products and local resources that are defined by the relationships between public and private actors. Farms are among the main actors of this phenomenon and the tourism activities they carry out are based on the use of resources present in the territory.

Differences in how agritourism is conceived and defined by how governments and policymakers are treated by taxing and regulation authorities. If the definitions are too vague, they can result in an erosion of overall tourism product quality. However, if the definitions are too restrictive, they can result in being too elitist or too small to matter [41]. In response to the various definitions, a team of US researchers created a framework definition of agritourism for which the core of activities is deeply connected to agriculture and takes place on a working farm [42].

The first national law to define agritourism was issued in Italy in 1985 (no. 730/1985). It deals with the diversification of income sources for working farms in rural areas through overnight stays. A more restrictive national law (no. 96/2006) established that agritourism can only be performed by the farmer and their family members, and agricultural activity of the farm must be predominant in terms of working hours and not in terms of income.

The rationale of the law pursues goals related to economic issues (integrating the income of the farmers and promoting the local products), social and cultural issues (enhancing the relation between the countryside and the city as well as preserving the local traditions), environmental issues (protecting the environment and the landscape), and occupational issues (creating new job opportunities with the aim of improving the quality of life and limiting the exodus from rural areas) [3].

Finally, in the development of multifunctionality in Italy, favored by the innovations introduced with Legislative Decree no. 228 of 2001 (the so-called Orientation and Modernization Law of the agricultural sector), the paths attributable to the short supply chain over the years have been at the center of important innovations and organizational efforts by farms, leading to an affirmation of the phenomenon of direct sales management of agri-food products and allowing them to be better valued in terms of equity wages for the agricultural system with high levels of consumer satisfaction.

Farms are now well integrated, structured, and consolidated in the farmers' markets [43].

In 2019, 17% of agricultural producers resorted to some form of direct sales, allocating on average 73% of their production. The number of producers practicing direct sales in the first half of 2020 has grown to almost 22% (+5% compared to the same period in 2019). Overall, the average share of production distributed through the direct sales channel in the first half of 2020 was almost 19% (+6.4%). Direct selling thus becomes the third distribution channel used by farmers, after being transferred to cooperatives, consortia, and producer organizations, and selling to wholesalers and commercial intermediaries [11].

It is therefore important to underline that, in addition to the negative effects of the pandemic in progress, the agricultural sector has demonstrated a further capacity for direct sales in recent months, both for reorganization and to seize some new opportunities. Furthermore, from the Ismea survey [44], it emerges that agritourism businesses use the direct sales channel even more than other agricultural businesses, favoring it clearly over other product outlets.

## 3. Materials and Methods

*Data Collection*

The analysis which has been carried out in the context of this study draws from different data and information sources.

With the aim to document the phenomenon crisis in a context of rapidly emerging information based on ad hoc reports of leading players and various agricultural organizations in the food sector, an extensive desk research and analysis of academic literature on the impact of COVID-19, including that disseminated by public and private research bodies relating to entrepreneurship, in the agri-food sector has been conducted.

The methodological approach which was chosen consisted of a systematic review of the existing knowledge by investigating both the theoretical and empirical information aimed to focus on the timely academic and policy debate regarding new trends and challenges on the agri-food sector in crisis times.

The empirical data for this study were collected using a semi-structured questionnaire, managed by the research team. The survey was conducted online, compatible for both computer and handheld devices, in summer 2021 (from 24 May to 31 July).

Due to the COVID-19 pandemic, all the data collection was conducted online. The survey distribution approach was designed using a random stratified sampling that ensured a good coverage of the national territory.

On the basis of the research purposes, the survey panel was constructed considering only farms with profitable support and secondary activities, i.e., those which, in addition to carrying out agricultural activities, are engaged in direct sales, in the transformation of products, and in the supply of catering and hospitality services. In accordance with the definition provided by ISTAT [45], the main activities under investigation are shown below Table 1.

**Table 1.** Activities connected with agriculture [45].

| Support Activities | Secondary Activities |
|---|---|
| <ul><li>subcontracting</li><li>first processing of agricultural products</li><li>seed processing for sowing</li><li>new crops and plantations</li><li>maintenance of the land in order to keep it in good agricultural and ecological conditions</li><li>services for livestock breeding</li><li>arrangement of parks and gardens</li><li>forestry activities</li></ul> | <ul><li>farmhouse</li><li>health, social and educational services</li><li>tourism, hospitality, and other free time activities (company visits, sports)</li><li>wood processing</li><li>crafts</li><li>renewable energy (photovoltaic, biogas, biomass) destined for the market</li><li>processing of vegetable (fruit) and animal (meat, milk) products and direct sales</li><li>aquaculture</li></ul> |

Following the methodological indications proposed by ISTAT [45], the activities related to agriculture have been divided into two categories: the category relating to support activities, not aimed at harvesting agricultural products, which leads the farm to differentiate its agricultural potential, and the category related to secondary activities. The latter are aimed at a process of broadening the functions of the business that produce income, some of which may also be completely independent from agricultural production.

Furthermore, the sampled farms had to be located at least 50 km of the relevant provincial capitals of reference.

To ensure adequate national coverage, the scouting activity of the farms was carried out on some specialized websites. Most of the contacts were provided by the: (i) Green tourism (www.turismoverde.it); (ii) Agritourism (www.agriturist.it); (iii) Terra nostra (www.campagnamica.it/la-nostra-rete/gli-agriturismo/).

The lists of farms were subsequently checked by the researchers to exclude farms that have ceased, those that have suspended their activity or, more simply, that had updated their contact details (e-mail and telephone number).

The sample contains 441 farms in total, distributed equally throughout the whole Italian territory. In order to ensure the same level of representativeness within the national territory in the sample, an average of 5% of farms with agritourism were extracted for each Italian region, also taking into account any reserves for non-responders.

Participants completed the online survey upon invitation. The average duration for completing the questionnaire was approximately 15 min. A total of 179 questionnaires were received and after reviewing and excluding invalid sample questionnaires with deficiencies and errors, a total of 77 complete and valid questionnaires remained, for an effective response rate of 17.5%.

In terms of territorial coverage, the presence of at least one respondent per region is highlighted; however, not any farms took part in the investigation in the case of Emilia-Romagna, Molise, Sardinia, and Sicily. All the sampled farms were contacted twice by the researchers.

The survey was pre-tested with 2 participants, including a farm that had participated in a previous online survey [46].

The questionnaire contained 26 questions grouped into three sections: (A) farm structure (questions 1–12); (B) direct and indirect economic effects of COVID-19 (13–19); (C) crisis management strategies and future prospect (20–26).

As this is an online survey, the researchers favored the use of semi-structured questions to facilitate the respondent, while leaving the participants to add comments and to provide any opinions.

To estimate the direct and indirect effects generated by the pandemic on business, the participants were asked to compare the various variables under investigation (i.e., direct sales, agritourism services, product transformation) before and during the COVID-19 pan-

demic. The time frame for comparison was the year 2019. In these cases, the questionnaire contained a 5-value scale ranging from "decreased" to "increased considerably".

Participants in the survey were also asked to provide forecasts, based on the observation of their business realities, on possible future scenarios (section B), in terms of:

1. changes in consumer demand for food and farm response;
2. acquisition of new customers;
3. changes in customer purchasing and consumption behaviors;
4. revenue forecasts;
5. changes to the organizational structure;
6. expansion of the services offered.

Finally, questions were asked regarding the type of agricultural activity carried out (UAA, sector of specialization, possible breeding, etc.), and the characteristics of the farmer such as, for example, the degree, the year of start of the activity, and previous work experience (section A).

## 4. Results: Consequences of COVID-19

### 4.1. Main Characteristics of Agricultural Farms

The sample of farms that collaborated in the survey is distributed throughout the whole national territory; however, there is a greater participation in the survey by farms in the central-northern regions which represent 75% of respondents.

The age of the entrepreneurs highlights a clear belonging to the population classes over 40 years: 41–50 (32%), 51–60 (29%), 60 and more (19%). The reading of the age by geographical area also highlights in Southern Italy the highest percentage of young farmers aged between 20 and 30 years (7%), as well as a strong disparity in the presence of farmers over 61 among the territorial divisions considered (Figure 1).

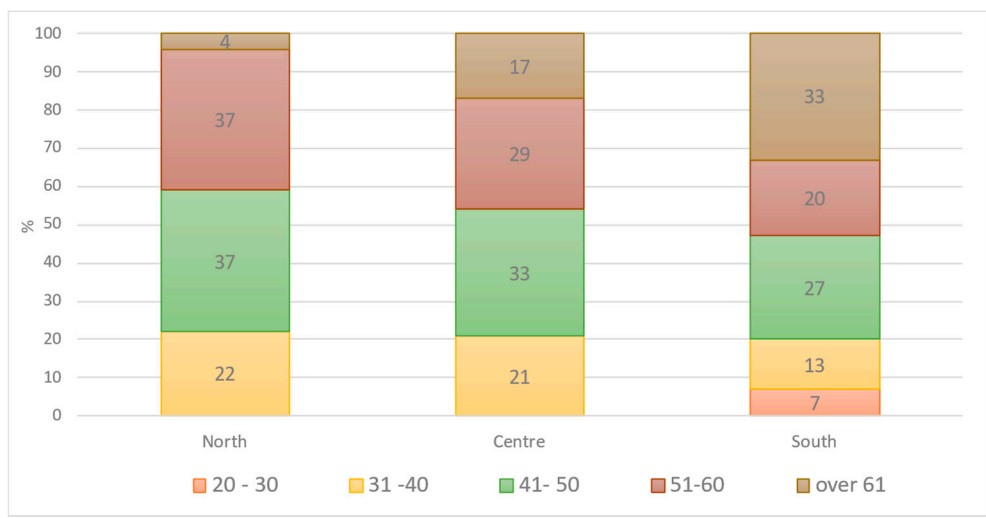

**Figure 1.** Distribution of farmers by age and territorial breakdown (values in %) (author: our elaboration).

The gender analysis shows a rather balanced relationship between men (51%) and women (49%). Additionally, in this case, the territorial focus returns a more accurate photograph of the phenomenon, showing a greater presence of female farmers in the South and in the North of the country where they amount to 35% and 21%, respectively, against 25% and 19% of male farmers. In the center, female farmers amount to 13% while male farmers amount to 22%.

Overall, 68% of respondents declared that they had carried out another work activity before starting the farm, but only in 6% of cases does previous work refer to the agricultural sector. The presence of heterogeneous professionals reveals an important radical shift in the Italian agricultural sector by almost all of the interviewees, a passage that leads us to believe—as already underlined in another work [46]—to be in the presence of a renewal of

the sector thanks to the entry of subjects with different experiences and skills. In addition, only 10% of the sample carried out entrepreneurial activity on a part-time basis, dedicating themselves to a job in another sector. This choice could find its justification in the need to guarantee income integration.

In line with the findings at national level [45], the survey sample has a medium-high school education. Indeed, 44% of the farmers interviewed declared they have a high school diploma out of which 16% have a diploma in agriculture. Graduated farmers are 33% of the total and 29% of them have obtained a degree in agriculture.

It is noted that the farms that participated in the survey are now quite consolidated, having started the agritourism business, or in any case a business diversification process, on average for 15 years. The reading by geographical area shows how the youngest farms in the sample, with an average age of around 12 years, are located in the Northern regions; followed by those of the Center with an intermediate age of around 17 years and the more mature ones with an average age of 19 in the South. Presumably, the farms that have matured a farm life cycle of more than fifteen years of activity are also those ones that have been able to better resist the crisis generated by the health emergency [13].

Moving on to consider some physical data on the structures involved in the survey, the analysis shows that 24% of the farms in the sample have an UAA between 5 and 9.99 hectares, a figure substantially in line with the national average [47]; among these, 11% are located in the North (Table 2). Farms with a UAA of between 10 and 19.9 hectares (17%) follow; in this case there is a concentration of them in the South. Finally, the large farms with a UAA greater than 100 hectares are 13% (in 6% of cases they are concentrated in the South). The small percentage of farms with a UAA of less than 5 hectares in all territorial districts, evidence that diversification processes are more frequent in farms that can count on a larger size.

**Table 2.** Utilized agricultural area by territorial breakdown (values in %) (author: our elaboration).

|        | 1–1.99 | 2–4.99 | 5–9.99 | 10–19.99 | 20–29.99 | 30–49.99 | 50–99.99 | 100 and More | DK/NO |
|--------|--------|--------|--------|----------|----------|----------|----------|--------------|-------|
| North  | 4      | 8      | 11     | 5        | 1        | 3        | 3        | 3            | 1     |
| Centre | 4      | 6      | 7      | 3        | 4        | 0        | 6        | 4            | 1     |
| South  | 0      | 0      | 6      | 8        | 1        | 1        | 3        | 6            | 0     |
| Total  | 8      | 14     | 24     | 17       | 7        | 4        | 11       | 10           | 3     |

The prevalent production orientation of the sample is 30% represented by permanent crops, which, together with arable crops (25%), largely affects farms in Central Italy. Vegetables, which account for 26% of the sample, are the prevailing orientation in farms in the South, while livestock, with 14%, is the prevailing orientation in farms in the North (Figure 2).

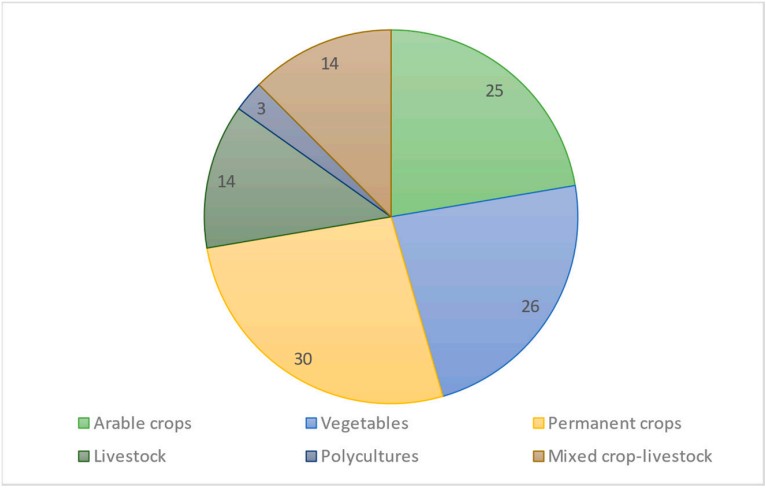

**Figure 2.** Agricultural land use in the sample farms (values in %) (author: our elaboration).

The farms in the sample analyzed produce mainly using sustainable methods: 56%, indeed, practice organic farming. Otherwise, the analysis of response rates does not reveal a particular attention to certified quality production, considering that only 22% of farms produce protected designation of origin (PDO) products, 19% protected geographical indication (PGI), and traditional specialties guaranteed (TSG) products, and 8% Denominazione di Origine Controllata/Denominazione di Origine Controllata e Garantita (DOC/DOCG) products.

The workforce involved in agricultural activity for more than 60 days a year is mainly made up of family members of the farmer, although almost six out of ten agritourism companies (58%) also see the participation of external collaborators.

In the farms of the survey sample, the most frequent connected activities are the first tillage and maintenance of the land, respectively, with 42% and 31%. As regards secondary activities, on the other hand, agritourism activities emerge with an incidence of 97% and direct sales, respectively, with a share of 65%, followed by the transformation of products (50%) and recreational and social activities (26%) (Figure 3).

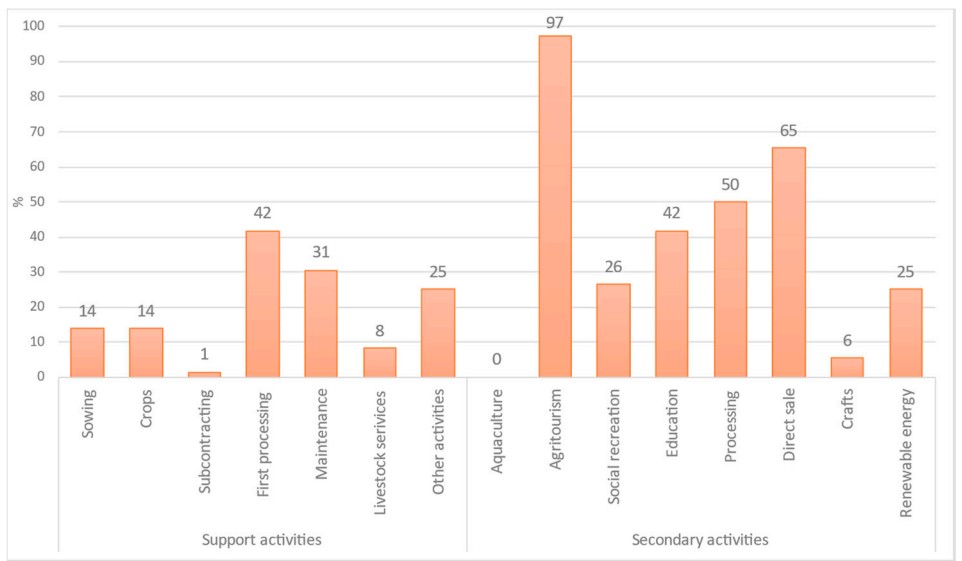

**Figure 3.** Support and secondary activities in the sample farms (author: our elaboration).

### 4.2. Perceived Impacts on Sales Channels and the Tourism Offer

The COVID-19 pandemic in Italy has had negative repercussions on the main production sectors of the country, primarily due to a general reduction in consumption. The adoption of a series of measures including social distancing, isolation, restriction of mobility, and closure of national borders undertaken by the Italian government to limit the spread of the virus contributed to supporting this effect. These are measures that have affected all phases of the food supply chain, determining a strong impact on food distribution [48], on the structure of demand [3,49,50] and, consequently, on entrepreneurial activity (Figure 4).

The agricultural and agri-food sector, despite having shown greater resilience than other productive sectors during the first phase of the pandemic, nevertheless had to face the negative effects resulting from its spread. Beyond the seriousness of the pandemic, the impact varied from farm to farm according to the resilience and ability of operators to adapt to the new demand structure, the interruption on the supply side, the additional costs required to adapt to COVID-19 measures, the government support measures, and their effectiveness [51].

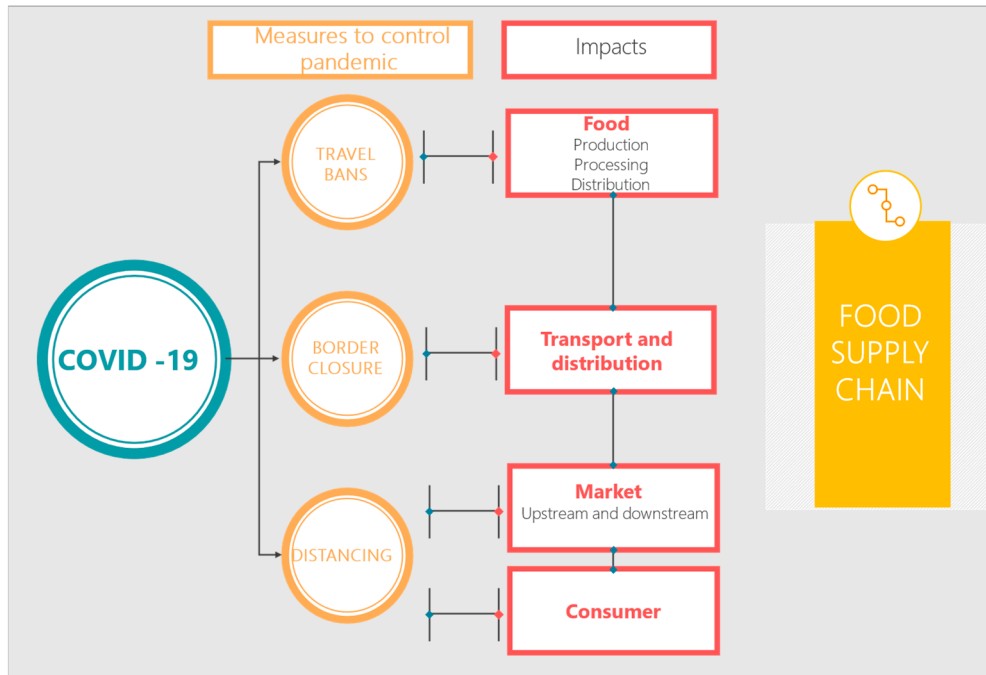

**Figure 4.** Impact of COVID-19 containment measures on the food chain (our adaptation from Poudel et al., 2020 [48]).

In general, the farm, although not having been subjected to work stoppages during the lockdown, has seen, in the short term, to reduce its sources of income deriving from support and secondary activities by having to give up, for a rather long period of time, the related financial revenues. Moreover, after some time, the same activities are still negatively affected by the changes in social behavior following the pandemic and which also affect demand and consumption styles (for example, the transition to greater domestic consumption), as well as on methods of tourist use.

The results of the survey show, a year and a half after the beginning of the pandemic, a general decline in the volume of activity witnessed by the change in turnover. All respondents reported the continuing downsizing of the overall gross turnover class deriving from agricultural and secondary activities. From the responses collected, it emerges that an increase in the subjects declaring a turnover of less than 20 thousand euros, a class which went from 10% to 34% due to the restrictive measures imposed by the lockdown, corresponds to a thinning of the other ranges considered, particularly significant in the central turnover class (40–60 thousand euros), which is reduced by 10 percentage points, and for a turnover above 80 thousand euros (Figure 5). In the latter case, the depressive effect of the pandemic affects almost half of the sample (47% of subjects).

These criticalities are directly attributable to the slowdown in the general economic system which, on the one hand, has invested to a lesser extent some components of the primary sector (with the exception of horticulture, the supply chains linked to the Ho.Re.Ca. channel and to exports have been subjected to production stoppages). On the other hand, it severely hit secondary activities, including direct sales, agritourism, and other forms of public reception on farms (hospitality, catering, educational farms, summer camps).

As a demonstration of the fact that the shock caused by the pandemic had a significant impact on the balance sheets of the agritourism farms and for a prolonged period of time, only 9% of the sample declared that they have not reported changes in overall revenues. Of these, 57% indicated substantial stability and almost 30% indicated a slight increase. On the other hand, among those who reported a decrease in revenues, it is highlighted that a decrease of more than 50% was suffered by a third of the sample, echoed with 24% of the share of those who declared a contraction between 30 and 50%. The incidence exceeds

40% of respondents in the case of variations of lesser intensity (incidence between 10% and 30%).

**Figure 5.** Incidence of gross turnover class (values in %) (author: our elaboration).

The effect of the virus containment measures, such as social distancing and the restriction of mobility, have affected the choices of consumers with respect to some sales channels used by farms (Table 3). In mid-2021, at the national level, a decline remains in the sale of agricultural and agri-food products, in the direct sales channels on the farm (11%), in local markets (6%), to wholesalers (8%), and catering (6%). However, at the level of geographic areas there are different behaviors: while in the North and Center the decrease is concentrated in direct sales within the farm, in the local market and in catering, in the farms of Southern Italy the reduction also extends to sales to wholesalers, consortia, and cooperatives.

**Table 3.** Impact of COVID on farm sales channels (values in %) (author: our elaboration).

| Sales Channels | Pre COVID | Post COVID |
|---|---|---|
| On-farm consumption * | 47 | 48 |
| Direct sales on the farm | 42 | 31 |
| Direct sale outside the farm | 25 | 19 |
| Sales to wholesalers/brokers | 48 | 40 |
| Large-scale retail sales | 5 | 6 |
| E-commerce-online sales | 19 | 25 |
| Sale to cooperatives/consortia/producer organizations | 45 | 48 |
| Specialized retailers/restaurants | 24 | 18 |
| Other | 38 | 57 |

* part of agricultural production not destined for the market, but for the internal consumption of the family or for the farm in order to create methods of income integration linked to the organization of human, productive and financial resources and to the internalization of their use [52].

As a response to the contraction of traditional sales channels, the use of e-commerce proved to be a valid support for the trade of farm products during the pandemic. This is true both for farms that had already activated online sales on their website or on dedicated apps and social channels, and for those that have resorted to it in response to changes in purchasing and consumption habits. The increase in the use of e-commerce for the farms in the sample is around 10%. These data confirm one of the characteristics of the pandemic, namely the transition from face-to-face relationships to digital ones [7,53] with the consequent increase in online sales [3]. Bakalis et al. [2] (p. 171) highlighted how the 'e-commerce shopping may become the norm for shopping, supporting reduction of any kind of transmitted viral diseases in places which are commonly crowded'.

The strategies for the placement of farm products also saw a strengthening of commercial relations with large-scale retail, as in the case of 11% of farms in the South, or by

resorting to the channel of wholesalers (11%) and intermediaries (13%), as in the case of farms in the Center and in the North.

It should be noted that the increase (19%) in the item 'Other' finds its justification in combining the delivery of ready meals with that of processed agricultural products not purchased online. The delivery of prepared products appears to be, on the one hand, a substitution of abruptly interrupted agritourism services, and on the other as a response to a demand and a supply system for suddenly changed food products [13]. 'Definitely online ordering and delivery are the most drastic change' [2] (p. 171).

Self-consumption marks a negligible change on a national level. However, at the level of territorial distribution, it is distinguished by an increase of 35% in the farms in the North and 29% in those in the Center. Considering the climate of general uncertainty and the decline in the performance of traditional sales channels, it is possible to believe that the increase in self-consumption is due to the farm's decision to strengthen the quantity of products destined for the family's internal consumption and online sales channels, large-scale retail, and that of other commercial intermediaries. It is undeniable that the crisis resulting from the spread of COVID-19 has created a difficult environment, but at the same time new business opportunities have also been generated in the agri-food sector [5], as the results of the survey show.

The restrictions on sales to the public as well as the discontinuation of the Ho.Re.Ca. have resulted in changes in the business structure, which do not exclusively concern agricultural activity tout court, but which directly affect the support and secondary activities.

If, on the one hand, a negligible decrease is recorded for the item 'work on behalf of third parties' (8%), on the other, the activities that struggle the most are direct sales and the transformation of products for which a decrease is perceived, respectively, by 39% and 25% (Figure 6). At the same time, probably thanks to the search for new business strategies and/or the greater proximity to inhabited centers for many of the farms in the sample, direct sales and transformation have also seen increases although prudential feelings prevail among farmers. Less radical, however, are the changes in agricultural activity, which have remained stable by 68%.

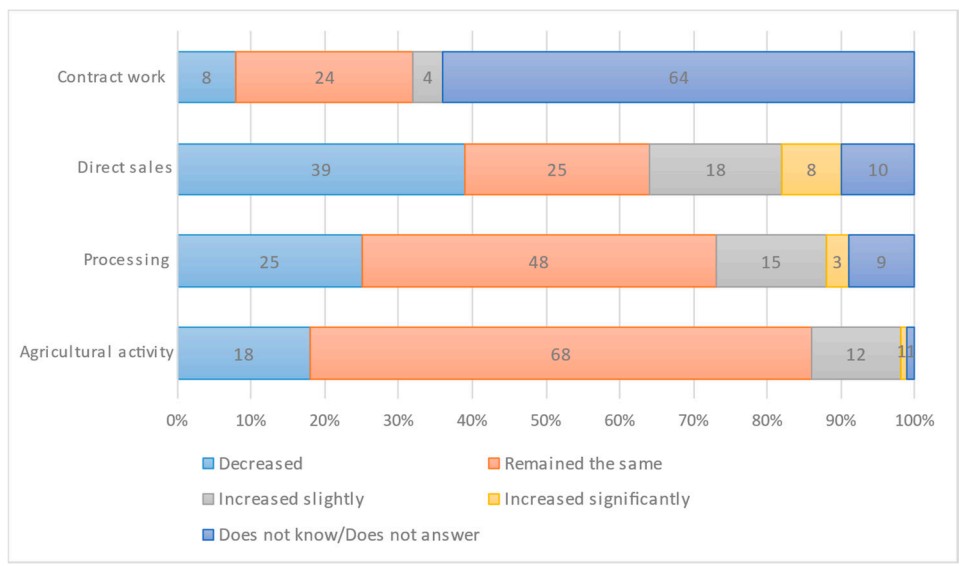

**Figure 6.** Perception of changes for support and secondary activities (values in %) (author: our elaboration).

While bearing in mind some structural differences in the sample, it is clear that the travel restrictions, as well as those imposed on tourism activities, have resulted in a penalty for agritourism farms, exposing them, among other things, to an increase in liquidity risk.

More specifically, the reception and catering activities were closed in spring 2020 but it is possible to say that the economic difficulties continued throughout the year: new closures were imposed in the autumn and winter months, with a consequent decrease in

flows domestic and foreign tourism. To this, the increase in costs must be added to comply with the provisions on safety.

As it was easy to hypothesize, the subjects of the sample recorded a decrease in all activities related to tourism and, more generally, to hospitality services and use of free time (Figure 7). The most penalized activities are those associated with food and wine tourism for which a 68% decrease was indicated, recreational and sports activities (66% decrease), educational farms, and educational services (64% decrease). At the same time, changes in the tourist and agritourism demand side have led to an increase, albeit slight, in outdoor activities, such as those related to natural (13%) and gastronomic (10%) tourism.

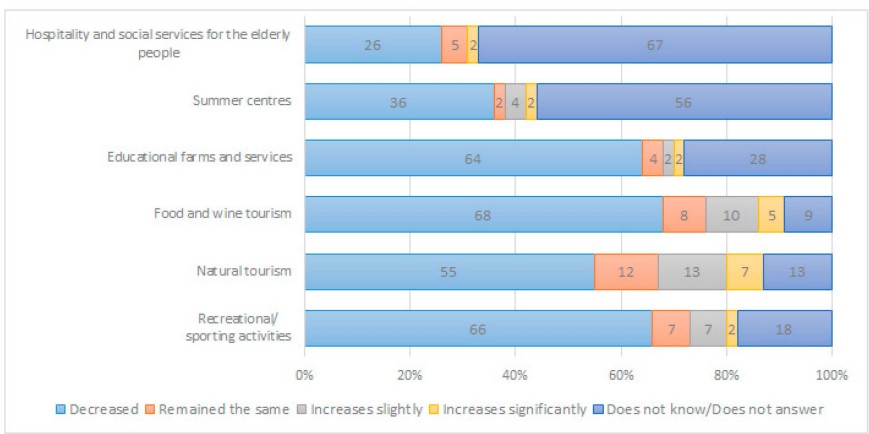

**Figure 7.** Perception of changes in the main tourism and hospitality activities (values in %) (author: our elaboration).

In comparison with 2019, there is a decrease in direct sales, both of that carried out in the farm (to the extent of 53%) and that carried out outside (41%) through the various commercial channels (Figure 8). However, agritourism services are the ones most penalized by the COVID-19 emergency. More than two thirds of the sample, indeed, reported a reduction; a similar situation concerns the drop in demand for secondary activities (63%).

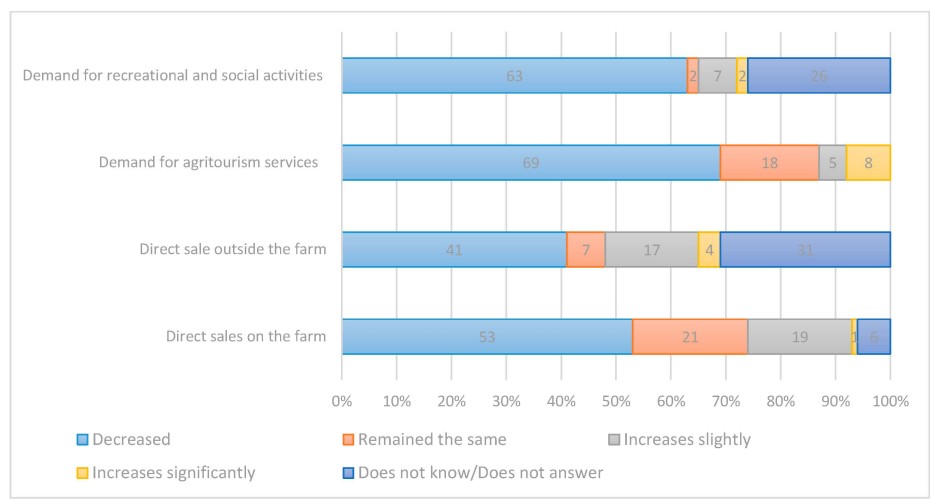

**Figure 8.** Effects on demand observed in the farm compared to 2019 (values in %) (author: our elaboration).

At the same time, in addition to situations of stability, farms declared they had a greater demand for products or services, highlighting a certain importance of diversified farm systems. Those positive changes could be mainly attributed to the proximity demand which led to increases, albeit marginal, both for agritourism services, recreational and social activities (12%) and, in particular, for the demand for agricultural and processed products (36%).

*4.3. Repositioning Strategies*

The impact of the COVID-19 pandemic on the activity levels of agritourism farms was sudden and with profound economic repercussions. The spread of the virus has stimulated new buying and consumption habits, some of which, as highlighted in some works [54,55], will also be maintained in the future.

The provisions aimed at containing the pandemic have led to attitudes that go in the direction of greater attention to health, hygiene, outdoors, well-being. The possibility of combining these needs with those of the holiday represents, in the direct experience of the interviewees, the main motivation (29%) that supports the post-COVID agritourism demand. The increased interest in quality productions is therefore towards a healthier food, from the point of view of food system resilience as defined by Tendalla et al. [56], and the renewed attention to environmental issues, together with the possibility of using them, represent almost half (48%) of the reasons that would push customers to choose the structures and/or products of the agritourism. This incidence is confirmed by research showing that a significant number of consumers have switched to purchasing healthier and more sustainable food, despite price volatility and uncertainty about the future [1,57,58]. There is also, according to the respondents, a component that is traced back to the preference for a direct producer–customer relationship.

This element would also seem to emerge from the direct survey, as evidenced by the acquisition of new customers (46% of cases) by the agritourism structures due to the changes that have taken place in the last year and a half. More specifically, 37% of new customers (Table 4) concern farm products and, to a greater extent, the demand for agritourism services (44%); customers who turned to agritourism even before the emergency showed greater stability in the case of demand for farm products (35%), even if at the same time there was a decrease (20%) and an increase to a negligible extent (8%).

**Table 4.** Type of customers and changes in purchasing and consumption processes (values in %) (author: our elaboration).

| Agritourism Services | Values in % | Agricultural Products | Values in % |
|---|---|---|---|
| New customers were acquired | 44 | New customers were acquired | 37 |
| Established customers increased demand | 20 | Established customers increased demand | 8 |
| Established customers decreased demand | 18 | Established customers decreased demand | 20 |
| Established customers left demand unchanged | 18 | Established customers left demand unchanged | 35 |

However, despite the critical elements, it can be argued that the pandemic has offered farms the opportunity to explore new opportunities, highlighting a strong capacity to react to exogenous shocks.

Half of the farmers (53%) decided to relaunch the agritourism by pursuing new business strategies. Ten percent of the sample clearly explained the construction of alternative sales channels to the traditional ones, and in 11% of cases the propensity to review production aspects emerges through an expansion of the offer of new products and/or services, to better respond to the changing needs of customers. The need to adapt to an increasingly variable context, also characterized by various limitations, has led some farms to modify the production, management, and organizational structure by closing, for example, the agritourism and related activities to focus on home delivery (5%); in addition, 21% of respondents expressed a sense of mistrust and concern about suspending the activity.

The centrality of the agricultural component remains a strategic and survival element of the agritourism business [13], but it will be necessary to redesign one's growth path to ensure the development of the farm.

The need to reposition themselves at the market level leads 87% of farms to foresee the activation of other forms of business diversification (Figure 9). Among these, hospitality

and direct sales prevail in the share of 18%, followed by the transformation of products (14%) and recreational, sports, and cultural activities (13%). The restrictions related to the protection and containment measures of the pandemic (such as, for example, separation of spaces, guarantee of spacing, sanitation) limit investments in the field of catering, an item that affects only 8% of the farms interviewed.

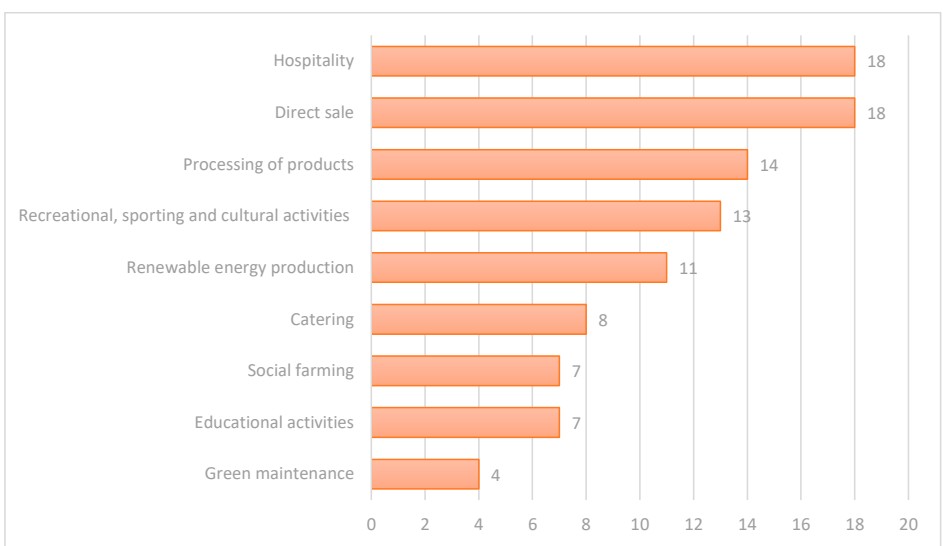

**Figure 9.** Forecast of new diversification activities to be implemented (values in %) (author: our elaboration).

With reference to the sale of products, three quarters of the sample declared that they activated new direct sales methods to respond to the changed demand for food. Among the new services offered, the home delivery of farm products stands out, involving almost half of the sample (45%). In response to a demand and a supply system for food products which have also suddenly changed due to the blocking of people movement and therefore of agritourism services, the farms responded with the conversion to a product delivery service, with all that ensued in organizational terms such as collection of orders, preparation of the delivery, and transport through a dedicated vehicle [13]. This context also includes the activation and use of e-commerce for one of four farms (24%) and the rapid spread of the home delivery service and ready meals, a service that also involved farmhouses.

The construction of alternative outlet markets to those traditionally used is interpreted in a differentiated way by the farmers of the sample (Figure 10). In addition to the fairly shared path of e-commerce, there is the adaptation of farms structures with the creation of spaces equipped for smart working (15%), with the aim of guaranteeing long-term hospitality for those who work remotely, but also didactic activities (13%). The evolution of the services offered also involved home delivery, in particular that of farm products (14%).

The demand for agritourism services by loyal guests remained stable compared to the pre-pandemic period for 18% of the sample and increased for 20% of them. However, a 18% share recorded the decrease in demand for services from established customers.

It is interesting to observe how the need to rethink their habits (i.e., favoring the forms of short supply chains) led consumers to turn their attention to the rural areas around the city of residence [59,60] in an area not exceeding 50 km. According to the sample, the new customers come from urban areas in 66% of cases for agricultural products and represent 88% of the demand for agritourism services (Figure 11), helping to increase the resilience of these territories and to create ecosystems richer in food but close to big cities. This requires, as indicated by Bakalis et al. [2], the adoption of coordinated actions between different actors.

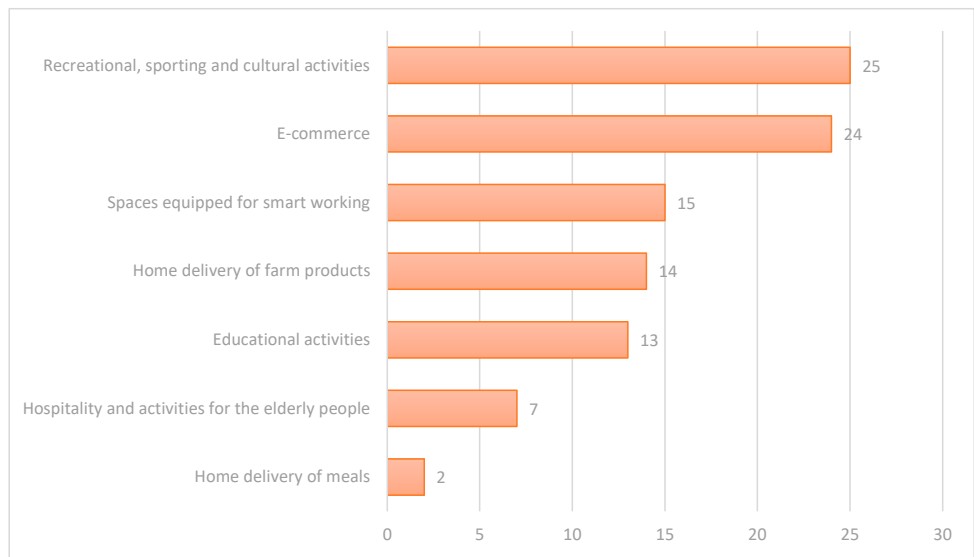

**Figure 10.** New business services in response to emerging needs (values in %) (author: our elaboration).

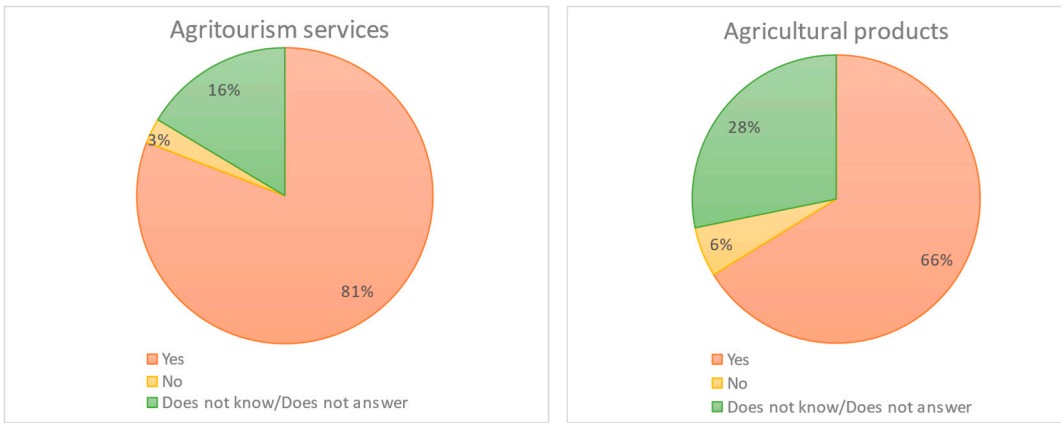

**Figure 11.** New customers: coming from urban areas (author: our elaboration).

Trust in the future unites the subjects of the sample who show a certain optimism with respect to the development of the business in the next two years: almost half of the farms (47%), indeed, indicate an increase in revenues (deriving from agricultural activity and related activities) and there is only a decrease in 16%. However, the share of those who declare themselves uncertain about what will happen weighs on the overall result, a sentiment that affects almost one of four interviewees.

## 5. Findings and Discussions

The COVID-19 pandemic has had profound effect on lives of the people and on business activities in the rural areas, which made it necessary to reconsider the business models [61] in terms of environmental, economic, and social dimensions. It has highlighted the changing role that food production and its distribution play in the economic development by creating opportunities for entrepreneurship in the agri-food sector with new digital technologies and innovations [5], allowing farmers to work better, even in crisis periods.

The present research analyzed the way the Italian farms with agritourism activities have responded to the crisis and explored how it has changed their business management while strengthening their resilience capacity.

The findings from this study show that farms with consolidated or long-term diversification strategies (started on average for 15 years) can boast better performances in economic

terms. In addition, as a response to the crisis, the sample of farms showed a greater orientation towards more conscious production models and food behaviors based on the trinomial health–environment–sustainability. The analysis of the responses and, in summary, all the information collected from the submission of the questionnaires, show a great reacting ability by farms and their willingness to make new strategic choices to respond to the new needs and requirements of the market and customers. On this aspect, the set of good cultural levels, experience, and diversified skills have played a positive role, as well as the awareness of their social and economic function in terms of valorization and care of the environment. Therefore, an important focus for future research should be aiming to better understanding the nature of farmer capabilities and how to strengthen them.

Moreover, the results show farms with agritourism activities have effectively faced the difficulties caused by the pandemic by expressing their ability to rapidly change their commercial channels and avoided disruptions to food production by guaranteeing food security and respecting the protocols and measures imposed by the crisis. While reporting the centrality of agricultural production activities, the survey showed that the impacts of the pandemic have had a fair degree of stability in secondary activities, and in particular on direct sales and product processing [3].

Among the most interesting features of the study, especially in terms of future strategies, we highlight the positive data on the forecasts for the relaunch and expansion of services and products that characterize the responses of farmers. An ever greater 'personalization' of products and services offered by agritourism facilities and a greater listening to consumers represent a renewed and more advanced form of care and strengthening of the fiduciary relationship, which also passes through digital forms. In addition, this is the evidence that during COVID 19, a significant number of consumers switched to purchasing healthier and more sustainable foods, and the crisis seems to have offered an unprecedented opportunity in the redesign [1] of the agri-food market by guiding the transition towards a more sustainable supply and production through the use of using alternative channels of sales (direct sales, home delivery, online sales). Consumers could make purchases directly from farmers and farmers had the opportunity to use digital technologies for selling their products. The enhancement of e-commerce, the new activities experienced in the pandemic crisis (i.e., food preparation, home delivery, expansion of guest services), represent new opportunities for farmers on which they intend to start new business paths.

It should be noted that COVID-19 has created a challenging environment for the agri-food sector and entrepreneurship, but at the same time, as our study highlights, new entrepreneurial opportunities have been generated to specialize and consolidate already-diversified strategies. This is a change that also raises a series of questions on the role that diversified farms, especially those with agritourism activities, can play on the future of the territorial areas in which they operate, placing themselves at the center of 'innovative ecosystems' in socio-economic and environmental terms [5].

In our opinion, from the analysis of the data collected, the political implications related to both the size of the farm and the role of diversified farms in the agri-food system clearly emerge.

Relationships emerge among the territories, communities, markets, citizens–consumers, and other farms. Therefore, agricultural production systems, community empowerment, and networking are identified as important determinants that characterize orientation of the Italian farms with agritourism activities. Further research also needs to be carried out by focusing on the cooperation among farmers, rural development organizations, and local communities aiming at unveiling the most urgent questions and offering reasonable ground to drive the envisaged sustainable food policy planning.

The research highlights the centrality of the agricultural activity and food production in terms of origin, safety, and quality. Therefore, it is important that for the design of the upcoming policies, a key role is assigned to support agriculture and food supply chains through a place-based approach. Farmer's market and direct sales need to be supported

as factors for increasing value and favoring the bargaining power in the supply chains towards farms and protecting consumers.

With A Farm to Fork Strategy [62], the European Commission is called to define a sustainable food system by the end of 2023 in which to promote a global approach to the role of sustainable food policy. This study urges institutions at a national, regional, and local level to create a broad debate on rural policies. It is necessary to identify resilience tools in the implementation of the Strategy, which consists of recognizing the differences and peculiarities of different farming systems with a focus on innovative practices to support market access for farms and to connect producers with consumers such as local e-commerce platforms or delivery services, even as a key to strengthening rural–urban connections [63].

It is worth acknowledging that the present study suffers from the limitations of the geographic area covered, as it has been conducted only in Italy. External validity of the results need to be confirmed in the light of a larger and representative sample of the population. Although food systems are not the same around the world, government responses have been similar in terms of lockdown and travel restrictions.

Furthermore, the fact that the pandemic was still ongoing at the time in which the present research was written, and in some instances, the lack of quantitative data allowing a deeper analysis of its impact on the sector, constitutes evident limitations. However, even if it does not allow an analysis of long terms trends, immediate insights into the impacts of the lockdown on the food supply chain have been provided.

The aim of the research is indeed to provide timely signals of the effects of crisis on diversified farms during the last three quarantine phase. Nonetheless, this work represents a starting point for further research to be carried out to better understand the main vulnerabilities of the EU agri-food chain and strengthen its capacity to respond to future crises. The challenge of the next years will be how to attend to these policy areas within the difficult context of a global economy.

**Author Contributions:** Conceptualization, B.Z., M.V., G.G. and F.L.; Formal Analysis, B.Z. and F.L.; Investigation, B.Z., M.V., G.G. and F.L.; Methodology, G.G. and F.L.; Data Curation, B.Z. and F.L.; Writing—Original Draft Preparation and editing, B.Z., M.V., G.G. and F.L. All authors have read and agreed to the published version of the manuscript.

**Funding:** This research received funding from the Italian Network for Rural Development 2014–2020, Fiche 2.1 "Eccellenze rurali", co-financed by European agricultural fund for rural development. This project is managed by the Italian Ministry of Agricultural, Agri-food and Forestry Policies (Mipaaf).

**Institutional Review Board Statement:** Not applicable.

**Informed Consent Statement:** Not applicable.

**Data Availability Statement:** More information and the full data can be requested from the authors of present work.

**Acknowledgments:** The authors thank the Italian Ministry of Agricultural, Agri-food and Forestry Policies for supporting this study and all the multifunctional farms that involved in the survey for providing the fundamental information basis of this work. The authors also thank Francesco Ambrosini for the valuable contribution in organizing the web survey.

**Conflicts of Interest:** The authors declare no conflict of interest.

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
