# Peer review of "Agritourism and Farms Diversification in Italy: What Have We Learnt from COVID-19?"

_land, doi:10.3390/land11081215_

Round 1
Reviewer 1 Report
I recommend it acceptence in current form.
Author Response
Thank you very much!
Reviewer 2 Report
1. It is well structured paper.
2. To have diversified agr. holdings one need everyday, regular flow of simple agr. foodstuffs in the regular channels. so it better if you change the name of article to effects of Covid-19 crisis to EU policy...etc...........
Author Response
DEar Sir,
thank you very much for revising our article.
The title has been revised as it follows: !Agritourism and farms diversification in Italy: what have we learnt from COVID-19?!
Best regards
Reviewer 3 Report
|
1. What is the main question addressed by the research?
The authors of the article take up important problems related to the functioning of farms in the time of the COVID-19 pandemic. Although the article is written in an interesting way, in my opinion, it is in it (both in the abstract and in the rest of the work) - Clearly formulated goals (the goals do exist, but are not explicitly stated). The sentence should begin with The aim of the presented research is ... - There is also a lack of clearly defined research hypotheses (hypotheses)
2. Do you consider the topic original or relevant in the field? Does it
The research issues presented in the article are topical and very interesting. It is also sufficiently encapsulated with literature reports by other authors.
3. What does it add to the subject area compared with other published
The presented article shows the strategies that can and should be implemented on farms in the time of a pandemic. The presented problems are supported by questionnaire surveys.
4. What specific improvements should the authors consider regarding the
In general, the research is carried out correctly, the number of repetitions is typical for this type of analysis. However, the lack of the use of advanced research methods, such as the grouping of farms according to typical features and the use of the ANOVA method, raises some dissatisfaction.
5. Are the conclusions consistent with the evidence and arguments
The conclusions are correctly formulated and result from the conducted analyzes.
6. Are the references appropriate?
The literature cited in the work corresponds to the current research status in the field of the discussed issues.
7. Please include any assitional comments on the tables and figures.
No critical remarks regarding tables and figures.
|

Author Response
Dear Sir,
thank you for revising our article.
Please find enclosed our reply to your comments.
Best regards

Reviewer 4 Report
Thank you for an interesting article. It is well written with a good research foundation entitling one to draw conclusions. The title could be more precise, now it is very general and does not correspond well with the content of the article.
Comments:
line 46 - needs a source for this statement.
line 183 and following: why so much space is devoted to agro-tourism and not to other activities,
line 265 - the sentence suggests that the farms should be located exactly 50 km from the city - or at least 50 km?
line 274/5 - what does equally distributed mean, can this be documented somehow in this article?
There is a lot of vague wording in the discussion that is difficult to interpret (e.g. lines 684-5, 687-690, 699+) that needs rewording. At present, they add little to the discussion.
line 771+ - again, a paragraph written in such a way that little follows from it.
Overall the survey is correct and conducted on a large group of farms. The statistics and their interpretations are useful in understanding the effects of the pandemic on farms and food chains. The discussion section and the summary, as it stands, do not correspond to the results section and therefore do not contribute much. After rewording and improvement of the discussion, the article is suitable for publication.
Good luck!
Author Response
Dear Sir,
thank you very much for your comments.
Please find enclosed our reply.
Best regards

Round 2
Reviewer 2 Report
I have no questions to the authors.